# Epitaxial Order Driven by Surface Corrugation: Quinquephenyl Crystals on a Cu(110)-(2×1)O Surface

**Roland Resel** [1],*, **Markus Koini** [1], **Jiri Novak** [2], **Steven Berkebile** [3], **Georg Koller** [3] and **Michael Ramsey** [2],*

1   Institut für Festkörperphysik, Technische Universität Graz, Petersgasse 16, 8010 Graz, Austria
2   Department of Condensed Matter Physics, Masaryk University, 611 37 Brno, Czech Republic
3   Institut für Physik, Karl-Franzens Universität Graz, Universitätsplatz 5, 8010 Graz, Austria
*   Correspondence: roland.resel@tugraz.at (R.R.); michael.ramsey@uni-graz.at (M.R.)

**Abstract:** A 30 nm thick quinquephenyl (5P) film was grown by molecular beam deposition on a Cu(110)(2×1)O single crystal surface. The thin film morphology was studied by light microscopy and atomic force microscopy and the crystallographic structure of the thin film was investigated by X-ray diffraction methods. The 5P molecules crystallise epitaxially with $(201)_{5P}$ parallel to the substrate surface $(110)_{Cu}$ and with their long molecular axes parallel to $[001]_{Cu}$. The observed epitaxial alignment cannot be explained by lattice matching calculations. Although a clear minimum in the lattice misfit exists, it is not adapted by the epitaxial growth of 5P crystals. Instead the formation of epitaxially oriented crystallites is determined by atomic corrugations of the substrate surface, such that the initially adsorbed 5P molecules fill with its rod-like shape the periodic grooves of the substrate. Subsequent crystal growth follows the orientation and alignment of the molecules taken within the initial growth stage.

**Keywords:** organic films; thin film epitaxy

## 1. Introduction

The molecule quinquephenyl (5P) belongs to the class of oligo-phenylenes, which are famous as stable blue emitters in organic light-emitting devices [1]. Quinquephenyl ($C_{30}H_{22}$) consists of five phenyl rings coupled via the para positions of the individual phenyl rings. 5P can be regarded as another model material, showing that molecular crystals can be ordered efficiently on single crystalline surfaces [2].

The current model of organic epitaxy is based on lattice matching considerations [3,4]. The development of ordered molecular layers on top of single crystalline surfaces is classified on the basis of the epitaxial matrix: molecular lattices can be commensurate or incommensurate, but they can also show a point-on-line or a line-on-line coincidence with the periodic substrate surface [4,5]. It has been shown that a point-on-line coincidence is connected with the minimum potential energy [6]. While molecular monolayers are frequently considered in connection with lattice matching, the applicability of lattice matching to epitaxially ordered crystallites is not well understood.

The epitaxial crystallisation of rod-like conjugated molecules like sexiphenyl, sexithiophene and pentacene on single crystalline surfaces are has been studied [7–9]. It is well known that the molecular organisation within the initial growth stage determines the orientation and alignment of the crystallites within thick films [10–14]. It turns out that lattice matching can explain one of the observed types of epitaxial crystallisation of sexiphenyl molecules on Au(111), while the other epitaxial orientations are determined by the initial adsorption sites of the molecules within the first monolayer [15].

The influence of substrate corrugations on the epitaxial crystallisation of conjugated rod-like molecules has been studied on several surfaces: TiO$_2$(110) [16], Cu(110)(2x1)O [17–20] and KCl(100) [21,22]. In most cases lattice matching cannot be used as an explanation for the epitaxial crystallisation, since a clear minimum of the lattice misfit does not appear [23]. This paper shows one specific example where lattice misfit sums predict a specific epitaxial order of a rod-like molecule but another quite different epitaxial order is observed experimentally.

## 2. Materials and Methods

The thin film was prepared in an ultra-high vacuum chamber (base pressure $< 1 \times 10^{-8}$ Pa), equipped with basic facilities to check the cleanliness and order of the inorganic substrate (Auger electron spectroscopy and low-energy electron diffraction). The Cu(110) surface was cleaned by repeated cycles of Ar$^+$-ion sputtering and annealed at 800 K. The oxygen layer was deposited on the clean copper surface by background exposure to 40 Langmuir of oxygen at 400 K. The oxygen induces a (2×1)O surface reconstruction with a rectangular lattice of a$_S$ = 5.102 Å and b$_S$ = 3.608 Å and $\gamma$ = 90° [24].

Para-quinquephenyl was purchased from Tokyo Chemical Industry (Tokyo, Japan) and used without any further purification. The material was evaporated by a Knudsen source and deposited directly on the Cu(110)(2×1)O substrate surface at room temperature using a deposition rate of 0.2 nm/min. The film thickness was monitored during the growth process by a quartz microbalance, and a final thickness of 30 nm was achieved.

The film morphology was investigated by light microscopy (LM) and atomic force microscopy (AFM). LM was performed with an BX51 microscope from Olympus (Tokyo, Japan) using back reflection technique. AFM data was acquired with a Multimode Nanoscope IIIa microscope from Digital Instruments (Santa Barbara, US). The system was operated in tapping mode using an advanced AFM tip for high-resolution imaging. The scanning velocity was low to obtain accurate results even for large island heights.

X-ray diffraction patterns were recorded with a X'Pert diffractometer (Philips, Almelo, The Netherlands) equipped with an ATC3 texture cradle and a flat graphite monochromator. For all measurements, CrK$\alpha$ radiation was used. The short length of the beam path (goniometer radius: 173 mm), together with the wide-open collimators and slits (primary collimator: $2 \times 1$ mm, receiving slit: 1.8 mm), increases the sensitivity of the system and reduces the measurement time. The sample was characterised by specular diffraction and the pole figure technique. Specular diffraction scans give the intensity distribution along the direction perpendicular to the sample surface as a function of the scattering vector q$_z$. Thus it allows the identification of crystallographic planes of the 5P crystals (and of the substrate), which are parallel to the sample surface. In contrast, for the pole figure measurement, the length of the scattering vector is kept constant but the direction of the vector is varied in the whole orientation space. The pole figure method allows us to determine the spatial distribution of poles (net plane normals) within the film [25]. A recent development allows the measurements of pole figures by using synchrotron radiation; grazing incidence X-ray diffraction is combined with rotating thin film samples [26].

The analysis of the experimental data is based on the crystal structure of 5P. The molecule crystallises in a layered herringbone structure within the space group P2$_1$/c and lattice constants of *a* = 8.07 Å, *b* = 5.581 Å, *c* = 22.056 Å and *ß* = 97.91° [27,28]. Diffraction patterns were calculated using the software POWDERCELL to obtain peak positions and peak intensities, the experimentally observed pole figures were simulated with the software STEREOPOLE [29,30].

Lattice matching calculations were performed to predict the epitaxial order of the 5P crystals. Two different approaches were used: first, a periodic potential calculation [31,32] was performed with the software EPICALC and, second, lattice misfit square sums were calculated [15,33].

## 3. Results and Discussion

Light microscopy of the 30-nm-thick film reveals a homogenous appearance of 5P islands over large areas. Elongated structures all arranged in the same direction are visible over a range of several hundred μm, as shown in Figure 1a. Investigations by AFM give identical morphology with larger magnification; the typical surface morphology of the 5P thin film is shown in Figure 1b. The individual islands are of elongated shape with a length of several μm, a typical width of about 500 nm and a height up to 80 nm.

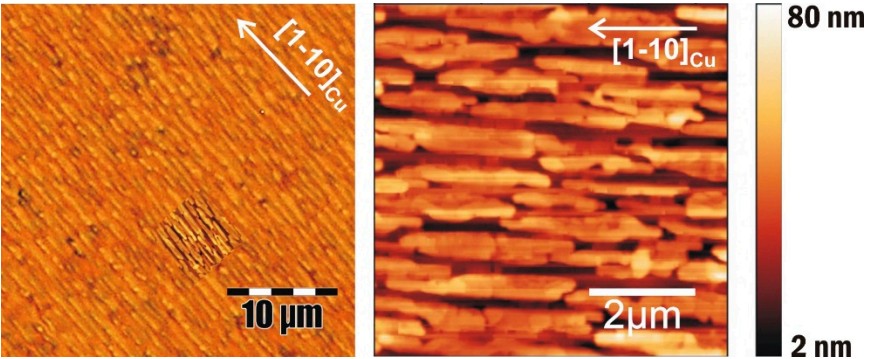

**Figure 1.** Microscopy images of a 30-nm-thick quinquephenyl film prepared on a Cu(110)-(2×1)O surface: light microscopy (**left**) and atomic force microscopy (**right**). The $[1{-}10]_{Cu}$ directions along the substrate surface are given by arrows.

The epitaxial alignment of the 5P crystals on the Cu(110)(2×1)O surface is performed using X-ray diffraction. Specular scans reveal only a single diffraction peak at 1.628 Å$^{-1}$, as shown in Figure 2a. A comparison with calculated diffraction pattern reveals that this peak belongs to the bulk crystal structure of 5P. The peak is indexed with 201 (calculated value: 1.636 Å$^{-1}$) and arises from the $(201)_{5P}$ net planes, which are oriented parallel to the substrate surface.

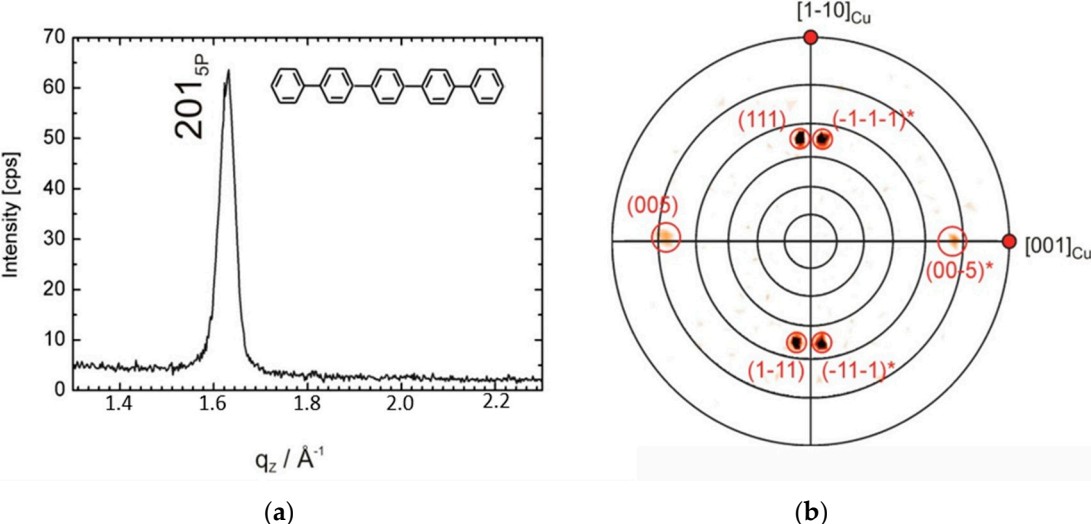

(**a**)        (**b**)

**Figure 2.** (**a**) X-ray diffraction (specular scan) of a 30-nm-thick quinquephenyl film grown on a Cu(110)-(2×1)O surface.; (**b**) pole figure taken at q = 1.4248 Å$^{-1}$ to find the spatial distribution of the 111 poles of the quinquephenyl crystals. Due to the finite resolution of the setup, 005 poles are also visible. Additionally, two directions on the substrate surface are given: $[1{-}10]_{Cu}$ and $[001]_{Cu}$. The inset in (**a**) shows the chemical structure of the molecule quinquephenyl.

In the next step, the in-plane alignment of the 5P crystallites is determined relative to the in-plane directions of the Cu(110)(2×1)O surface. For that reason, a series of pole figures is measured on the

high-intensity peaks of 5P together with a single pole figure of the 111 peak of Cu. Figure 2b shows one result of the pole figure measurements. The spatial distribution of the $111_{5P}$ poles is measured at $q_{111}$ = 1.4248 Å$^{-1}$. Four directions of enhanced pole densities are observed. In addition, two weaker poles originating from $d_{005}$ lattice spacing are observed due to the comparable reciprocal space vector ($G_{005}$ = 1.4380 Å$^{-1}$). Two distinct directions at the surface of the substrate—namely $[1–10]_{Cu}$ and $[001]_{Cu}$ – are determined from the $111_{Cu}$ pole figure and given within the pole figure.

From the XRD measurements, the crystallographic alignment of the 5P crystals could be clearly determined. While only one out-of-plane alignment of the 5P crystallites is present (($201)_{5P}$ parallel to $(110)Cu$), there are two different in-plane alignments of the crystallites: the pole densities from one specific alignment are marked by asterisks, while the pole densities of the second alignment are not marked. The two-fold symmetry of the substrate in-plane lattice allows for the formation of two equivalent orientations for each crystallite alignment, resulting in two enhanced pole densities for each.

The epitaxial relationships can be described by $\pm (201)_{5P} \parallel (110)_{Cu}$ and $\pm [010]_{5P} \parallel [1–10]_{Cu}$. The epitaxial matrix C that connects the unit cell vectors of both lattices is determined as follows:

$$C = \begin{pmatrix} 0 & \pm12.73 \\ \mp1.09 & 0 \end{pmatrix}.$$

The lack of an integer or rational numbers in a row or column of the matrix suggests incommensurable lattices of the substrate surface and the contact plane of the 5P crystals [4].

Lattice matching analysis is performed by using the surface unit cell of the Cu(110)(2×1)O substrate and the periodicity of the $(201)_{5P}$ plane. The lattice misfit is determined as a function of the in-plane angle $\Phi$ between the surface unit cell vectors of the substrate and that of the organic overlayer $a_{\parallel 5P}$. The two-dimensional lattice of the organic layer is derived for the crystal structure of 5P, with values of $a_{\parallel 5P}$ = 45.92 Å and $b_{\parallel 5P}$ = 5.581 Å and $\gamma$ = 90° taken from $a_{\parallel 5P}$ = $[10–2]_{5P}$ and $b_{\parallel 5P}$ = $[010]_{5P}$ [27]. The two-fold symmetry of the substrate requires a variation of the angle $\Phi$ only in the range between 0° and 180°. The squared misfit sums are performed for different extensions of the 5P lattice by varying the numbers of unit cells. Already at small extensions of the 5P lattice a minimum in the lattice misfit is obtained at $\Phi$ = 0° (corresponding to $\Phi$ = 180°); this minimum becomes pronounced and remains as a single minimum at larger extensions of the 5P lattice. Figure 3 shows a detail of the lattice mismatch calculations around $\Phi$ = 0°. Periodic potential calculations also predict a minimum at $\Phi$ = 0°. Since lattice misfit minima are considered as the minimum in the potential energy of two lattices [6], the epitaxial order of 5P on Cu(110)(2×1)O is expected at $\Phi$ = 0°. The epitaxial matrix $C_{calc}$ is calculated via

$$C_{calc} = \begin{pmatrix} -9 & 0 \\ 0 & +1.54 \end{pmatrix},$$

which represents a point-on-line coincident lattice [4]. However, the predicted epitaxial orientation is not experimentally observed. The existing in-plane alignment of the 5P crystallites is rotated by an angle of $\Delta\Phi$ = 90° to the theoretically predicted value so that the engaged epitaxial orientation is at $\Phi$ = 90°. To give an explanation for this inapplicability, the alignment of the individual molecules relative to the substrate surface has to be considered.

By using the epitaxial relationships together with knowledge about the arrangement of the 5P molecules within the crystallographic unit cell, the relative orientation of the molecules to the Cu(110)(2×1)O surface can be derived: the long axes of the molecules are parallel to $[001]_{Cu}$, i.e., parallel to the rows of the (2×1)O reconstruction, and the aromatic plane of the molecules is side-tilted by an angle of 36° relative to (110)Cu. This orientation of 5P molecules in the (201) contact plane allows only two types of epitaxial crystal alignment, which are observed experimentally. A similar situation was found for the growth of the molecule sexiphenyl on the Cu(110)(2×1)O surface: near-edge X-ray absorption fine structure spectroscopy (NEXAFS) revealed that the first monolayer molecules have

their long molecular axes along $[001]_{Cu}$ and a defined tilt angle of the aromatic planes relative to the substrate surface [18].

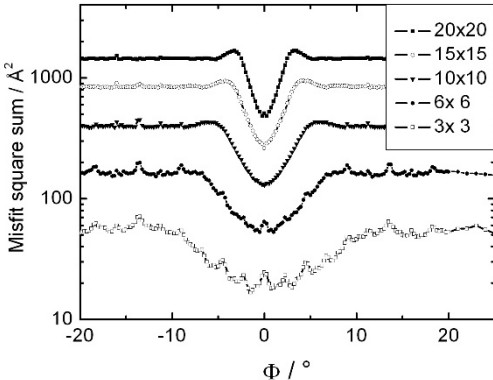

**Figure 3.** Lattice square misfit sums calculations of the quinquephenyl (201)5P lattice on the Cu(110)(2×1)O lattice for different rotation angles Φ. The extension of the quinquephenyl lattice varied from nine unit cells (3 × 3) to 400 (20 × 20).

From the analogy of the sexiphenyl growth on the oxygen-reconstructed Cu(110) surface we infer the following growth mechanism [18,34]. The Cu(110)(2×1)O surface provides a regularly corrugated surface with rows of oxygen formed along $[001]_{Cu}$ and a periodicity of $d_O = 5.102$ Å perpendicular to the oxygen rows [35]. This surface acts as a template for the alignment of individual 5P molecules: the molecules fill the substrate such that the long molecular axes are aligned along the oxygen rows and the aromatic planes are side-tilted relative to the substrate, as shown in Figure 4. As for oriented sexiphenyl molecules, the (201) contact plane of the 5P crystal does not match the periodicity ($d_O$) of the (2×1)O rows, with the molecular width being $d_{5P} = 6.6$ Å. However, the molecules can achieve commensurability in the first layer by a simple geometric rearrangement: The 5P molecules increase the tilt angle of the molecular plane with respect to (110)Cu, while keeping their normal distance constant [18,34]. A simple geometrical consideration gives a tilt angle of the aromatic planes as $\arccos(d_O/d_{5P}) = 39°$, which is quite close to the 36° present within the crystal structure. Within the initial growth stage, the molecules adapt the periodicity of the substrate surface and the subsequent crystal growth is determined by the lattice formed by the adsorbed molecules. The theoretically favoured epitaxial order is prevented, since large areas of the lattice would have to rearrange by rotation.

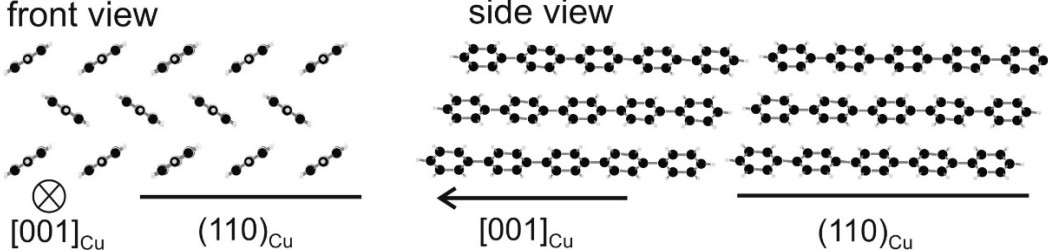

**Figure 4.** Real space view of the quinquephenyl bulk structure relative to the substrate surface and relative to the $[001]_{Cu}$ direction drawn in a front view and in a side view. The row of oxygen atoms on the Cu(110)(2×1)O surface are aligned along $[001]_{Cu}$.

In the case of inorganic materials, epitaxial growth is described by lattice matching of the overgrown lattice with the surface lattice of the substrate [36]. This concept is adapted for organic monolayers at single crystalline surfaces [37]. The concept of lattice misfit calculations is a tool that is used to predict epitaxial order for two-dimensional lattices (monolayers) as well as for three-dimensional lattices (crystals with preferred orientation) [32]. There are a few examples in the literature where epitaxial

crystallisation is explained by lattice misfit calculations [15,33,38]. The crystal growth of sexiphenyl on mica(001) as well as on Au(111) starts with a specific orientation and alignment of the molecules within the first monolayers; however, a rotation of only few degrees is required to adapt the lattice misfit minimum. Such rotations of crystal lattices from one to another epitaxial order are possible, as observed in the system sexiphenyl on KCl(100) [21]. Generally, it can be suggested that in the present case the rotation of the organic lattice is hindered, since the corrugation of the substrate surface and the (201) plane of 5P crystals complement each other topographically.

## 4. Conclusions

The epitaxial order of the molecule 5P on a defined Cu(110)(2×1)O surface is studied by atomic force microscopy and X-ray diffraction. A 30-nm-thick film shows islands with a length of several µm, a width of about 500 nm and a height of about 80 nm. The long edges of the islands are aligned perpendicular to the oxygen rows. The epitaxial alignment of the 5P crystals is experimentally observed, with the $(20\overline{1})_{5P}$ plane parallel to $(110)_{Cu}$ and $\pm[010]_{5P}$ directions parallel to $[1\overline{1}0]_{Cu}$. This implies that the long molecular axes are parallel to $[001]_{Cu}$ and the aromatic planes of the molecules are side-tilted to the substrate surface $(110)_{Cu}$.

Computation of lattice misfit square sums and periodic potential calculations show that there is a favourable alignment of the 5P lattice and the Cu(110)(2×1)O lattice and predict a point-on-line coincidence. However, the experimentally observed epitaxial order cannot be explained by lattice matching conditions. This leads to the conclusion that the epitaxial alignment cannot be predicted by the thermodynamic considerations of only the crystalline substrate and the organic overlayer lattices. In particular, the adsorption geometry of the molecules within the first monolayers has to be considered. In the present case, the surface topography of the oxygen-terminated Cu(110) surface acts as a template so that the 5P molecules arrange collectively by a specific crystallographic plane that fills the topography of the substrate surface.

**Author Contributions:** Conceptualisation, R.R. and M.R.; methodology, J.N. and G.K.; investigation, M.K.; and S.B.; writing—original draft preparation, R.R.

**Funding:** This research was funded by the Austrian Science Foundation (FWF): [P30222].

**Conflicts of Interest:** The authors declare no conflict of interest. The funders had no role in the design of the study; in the collection, analyses, or interpretation of data; in the writing of the manuscript, or in the decision to publish the results.

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
