# Peer review of "Epitaxial Order Driven by Surface Corrugation: Quinquephenyl Crystals on a Cu(110)-(2×1)O Surface"

_crystals, doi:10.3390/cryst9070373_

Round 1

Reviewer 1 Report

The authors report a structural study of the epitaxial growth of a model phenyl molecule on an oxidized metal surface. The results are of interest in the development of organic light emitting systems and appear to suggest that additional mechanisms for determining the crystallographic orientation of the molecular layers may be applicable. The x-ray scattering study in particular is carefully done and thoughtfully presented. The insights derived from the experimental work and supporting calculations are a useful and potentially important addition to the literature.

My opinion is that the work will merit publication once the authors have addressed several issues that I would like to bring to their attention:

1) My first (and most important) question regards the influence of the Cu/phenyl interface on the structure. In many systems (e.g. pentacene on metal surfaces) the first molecular monolayer deposited from the vapor can adopt a very different configuration from subsequent layers. Do the authors observe any change in structural parameters as a function of film thickness?  Do AFM studies indicate that the first layer has the same orientation as the thicker films probed in diffraction studies. If the authors have experimental insight then the paper should discuss it. Otherwise it would be very useful to have a statement of how the authors think the structural motif is propagated from the first monolayer through the relatively thick film.

2) The misfit sum presented in Fig. 3 considers an angular range near 0 degrees, but does not span the range discovered in the experimental results.  The authors should describe and if possible provide the misfit sum for phi=90 deg.

3) The “simple geometric consideration” giving a tilt of 39 deg. on p. 5 should be described with enough precision for readers to be able to repeat the calculation.

4) The caption for Fig. 2 should indicate that the pole figure is for {111} reflections of the phenyl molecule.

5) The stated goniometer radius of 173 cm seems too large for the angular resolution apparent in the results. Perhaps the authors intended a different value (or misplaced a decimal point)?

6) There are several typos and grammatical errors that should be corrected.  For example “famous as stable blue emitter” is missing an article and “orientation on the … surface is performed by x-ray diffraction” may be missing a verb indicating that the authors are determining the orientation using x-ray diffraction.

Reviewer 2 Report

Please find Detailed Comments and Suggestions in the attachment
